# Use of Mixed Microbial Cultures to Protect Recycled Concrete Surfaces: A Preliminary Study

**DOI:** 10.3390/ma14216545

**Published:** 2021-11-01

**Authors:** Lorena Serrano-González, Daniel Merino-Maldonado, Manuel Ignacio Guerra-Romero, Julia María Morán-Del Pozo, Paulo Costa Lemos, Alice Santos Pereira, Paulina Faria, Julia García-González, Andrés Juan-Valdés

**Affiliations:** 1Department of Engineering and Agricultural Sciences, School of Agricultural and Forest Engineering, University of León, Av. De Portugal 41, 24071 Leon, Spain; dmerm@unileon.es (D.M.-M.); ignacio.guerra@unileon.es (M.I.G.-R.); julia.moran@unileon.es (J.M.M.-D.P.); julia.garcia@unileon.es (J.G.-G.); andres.juan@unileon.es (A.J.-V.); 2Associated Laboratory for Green Chemistry-Chemistry and Technology Network (LAQV-REQUIMTE), Department of Chemistry, NOVA School of Science and Technology (FCT NOVA), NOVA University of Lisbon, 2829-516 Caparica, Portugal; pac@fct.unl.pt; 3Applied Molecular Biosciences Unit (UCIBIO), Department of Chemistry, NOVA School of Science and Technology (FCT NOVA), NOVA University of Lisbon, 2829-516 Caparica, Portugal; alice.pereira@fct.unl.pt; 4Civil Engineering Research and Innovation for Sustainability (CERIS), Department of Civil Engineering, NOVA School of Science and Technology (FCT NOVA), NOVA University of Lisbon, 2829-516 Caparica, Portugal; paulina.faria@fct.unl.pt

**Keywords:** construction and demolition waste (CDW), recycled concrete, surface treatment, polyhydroxyalkanoates (PHAs), mixed microbial cultures (MMCs), waterproof

## Abstract

One approach to tackle the problems created by the vast amounts of construction and demolition waste (CDW) generated worldwide while at the same time lengthening concrete durability and service life is to foster the use of recycled aggregate (RA) rather than natural aggregate (NA). This article discusses the use of polyhydroxyalkanoates (PHAs)-producing mixed microbial cultures (MMCs) to treat the surface of recycled concrete with a view to increase its resistance to water-mediated deterioration. The microorganisms were cultured in a minimal medium using waste pinewood bio-oil as a carbon source. Post-application variations in substrate permeability were determined with the water drop absorption and penetration by water under pressure tests. The significant reduction in water absorption recorded reveals that this bioproduct is a promising surface treatment for recycled concrete.

## 1. Introduction

The great increase of the population and consequent increase of urbanization and industrialization require the use of a large volume of concrete [1,2]. To meet these needs, the annual global concrete consumption is estimated to be about 25 billion tons [3]. It is forecasted that the volume of construction will continue to increase at an average growth rate of 3.9% per year [4]. The increased demand on concrete also implies the need of large amounts of its major components, cement, water and aggregates. Aggregates represent 60 to 75% of the total volume [5], which accounted for around 40 billion tons in 2015, and is expected to rise to 47.5 million tons by 2023 [6]. The worldwide consumption of concrete in the last decades has contributed to the depletion of non-renewable natural resources, usually extracted by digging mountains, mining and breaking river gravel [7], and has a serious negative environmental impact. Consequently, the development of eco-friendly alternatives is a challenge for the concrete industry in many territories [8,9].

In addition, the construction industry is responsible for the production of 50% of the waste generated worldwide [10], the management of which contributes significantly to the total costs. Therefore, a large part of this waste ends up in landfills [11], with consequent environmental problems such as contamination of soil, water and air and impacting human health [12]. About 8% of global CO_2_ emissions come from concrete production [13], causing global warming by trapping the radiant energy of the sun in the atmosphere [14]. Hence, the concrete industry is one of the main sources of massive global pollution that has a great impact on climate change [15,16,17].

As a result, there is a constant search for sustainable solutions, although sustainability requires a major change to the approach of the construction lifecycle, both short- and long-term, including building construction, maintenance, demolition and recycling [18,19].

However, it does not suffice to use recycled aggregates to reduce CDW stockpiling; the durability of structures and their service life must also be lengthened [20]. Concrete structures are expected to resist weathering, chemical attacks and the action of physical agents over their design lifetime while maintaining their properties intact [21,22]. Despite such expectations, several studies [23,24] have shown that concrete permeability and microcracking are related to the ingress of environmental ions, liquids and gases into the material that may directly or indirectly induce deterioration and shorten their service life.

One measure that may be adopted to enhance concrete durability is the use of surface treatments in situ, the application of coatings and impregnation treatments [25] that may be of utility in conjunction with new types of concretes, produced with innovative materials. Some examples are the use of apricot shell [26], graphene oxide [27], bio-deposition [28] or silane-based products [29]. These surface treatments aim to prevent deterioration and damage when exposed to extremely aggressive environments, and to further extend service life [21].

Most synthetic products, admixtures and (pozzolanic) additions, including fly ash or blast furnace slag, are associated with the adverse environmental impact of toxic emissions [30]. Such issues have driven research using biopolymers as alternative synthetic admixtures [31,32,33]. Nonetheless, the performance of biopolymers as concrete formulation admixtures or surface treatments in concrete bearing recycled aggregates has not been deeply studied.

Polyhydroxyalkanoates (PHAs) are biodegradable natural plastics [34] synthesized mostly by prokaryotic microorganisms, and are eco-friendly and renewable materials [35]. The microorganisms able to produce PHAs can do so during their growth phase or under nutrient limitation (N, P, Mg, K or S), while having an excess of carbon source [36,37]. PHAs are linear polyesters of R-hydroxyalkanoic acids stored by bacteria in their cytoplasm as carbon and energy reserves, among other functions. These inclusion bodies may account for up to 96% of the cell dry weight of the cultures [38]. PHAs are generally classified into three different classes according to the number of carbons of their monomers: short-chain length (3–5 carbons), medium-chain length (6–16 carbons) and long-chain length (more than 17 carbons). The simplest and most commonly used short-chain PHA is poly(3-hydroxybutyrate) or P(3HB). This homopolymer is brittle, presents high crystallinity and its suitability is restricted to few applications [39]. The presence of a second monomer in order to obtain a more elastic and flexible copolymer enlarges the number of utilizations of those polymers [40].

PHAs are particularly useful materials, given the breadth of their properties. Due to their production cost using pure cultures they have been applied primarily in the medical field, although many other uses also exist. At the moment a promising application is in food packaging. The versatility and other beneficial features PHAs have also highlighted the potential of these polymers for protecting concrete surfaces against the ingress of deleterious agents.

The present study was conducted to determine the capacity of bioproducts derived from polyhydroxyalkanoate-producing mixed microbial cultures to waterproof cement-based materials in order to reduce the ingress of aggressive agents.

## 2. Materials and Methods

### 2.1. Materials

#### 2.1.1. Bioproducts

The bioproducts used in this study to treat the surface of concrete specimens were generated by mixed microbial cultures (MMC). The culture medium contained tap water, ammonia as a nitrogen source and pinewood bio-oil obtained through the fast pyrolysis of waste pinewood (mostly the pyrolysis products of cellulose, hemicellulose and lignin contents) as a carbon source. The PHA-accumulating culture enrichment using pinewood bio-oil was performed in a reactor operated as a sequencing batch reactor (SBR) with a working volume of 1500 mL and operated under feast/famine conditions [41].

The PHAs present in these cultures were primarily of the short-chain length type, P(3HB) or P(3HB-3HV). Waste biomass from the reactor was added to water at a concentration of 2–10% to prepare two liquid bioproducts: SP (MMC sonicated to break down bacterial cell membranes) and NSP (MMC in which the suspension contained whole, non-sonicated cells). The SP bioproduct was subject to six sonication cycles, each with 3 min of ultrasonication followed by a 3 min recycle delay. Both obtained suspensions yielded a low-density, low-viscosity product readily applicable to concrete surfaces.

#### 2.1.2. Concrete

Bioproducts’ effectiveness was tested on recycled concrete specimens mixed as summarised in Table 1. Concrete components included EN 197-1-CEM III/A 42.5 N/SR blast furnace cement [42], natural river siliceous sand (0/4 mm) and as coarse aggregate a 50% blend of natural siliceous gravel (4/16 mm) and mixed recycled aggregate (MRA, 4/16 mm) supplied by a construction and demolition waste management plant (TEC-REC, Tecnología y Reciclado S.L., Madrid, Spain) (composition in Table 2). Both the fine and coarse aggregates were characterised prior to use to verify their compliance with the requirements laid down in Spanish structural concrete code EHE-08 [43] and European standard EN 12620 [44]. The recycled concrete was mixed for a water/cement ratio of 0.55 and a target strength of 25 MPa, which has previously been evaluated by García-González et al. [45].

To counteract the higher water sorptivity associated with recycled aggregate due to the presence of bound mortar on the surface and the nature of the masonry/fired clay fraction comprising it, the MRA was pre-soaked in keeping with industry practice for manufacturing recycled concrete suitable for applications not requiring high mechanical strength [46].

### 2.2. Concrete Specimens and Surface Treatment

Twenty-four 50 × 50 × 100 mm^3^ laboratory specimens were prepared, from three prismatic moulds of 100 × 400 × 100 mm^3^. The specimens were divided into three groups of eight, applying the non-sonicated (NSP) treatment to one group, the sonicated (SP) product to the second and water (H_2_O) to the reference group. After 28 days of curing, the bioproduct was applied in a 50 × 50 mm^2^ surface, not in contact with the mould. Two coats of treatments, at a concentration of 0.1 mL/cm^2^, were applied drop by drop with pipettes to better monitor the biofilm generated, to prevent bioproduct overflow and to ensure uniform distribution across the treated surface.

Nine 100 mm diameter and 200 mm high cylindrical specimens were also prepared similarly to the prismatic ones. A circular area of each of the three specimens was treated with NSP, SP and H_2_O, similarly to the prismatic specimens, again after 28 days of curing.

The aforementioned treatment yielded perceptible results in terms of concrete durability, and was sufficient to improve its durability. At the same time, the amount of bioproduct used did not induce the precipitation of particles from the MMC. The specimens were treated under the following environment conditions: 40 ± 5% relative humidity and 20 ± 2 °C.

### 2.3. Test Methods

The tests conducted to quantify the efficacy of the bioproducts for waterproofing concrete surfaces are described below.

#### 2.3.1. Water Drop Absorption in Hardened Concrete

Three days after the application of the treatments, water drop absorption was carried out based on the RILEM II 8b [47] test procedure, as previously schematized by Parracha et al. [48]. The 50 × 50 mm^2^ treated surfaces of specimens were water-drop-absorption tested by measuring the time required to absorb each of the nine applied droplets (50 µL/drop), to cover the entire surface without overlapping the water drops. The test was carried out under controlled environmental conditions inside a test room, with humidity and temperature conditions of 45 ± 5% and 20 ± 2 °C. The average of these measurements was adopted as a value for the entire treated area. The absorption times for the three sets of specimens (SP, NSP and H_2_O) were calculated as the difference between the exact (video-recorded) times when the water drop hit the concrete surface and when it was fully absorbed (Figure 1). The variation in the effect of the bioproduct (SP, NSP and H_2_O) on the treated surface over time was determined by conducting the test at 3 d, 7 d, 14 d, 21 d, 28 d, 42 d, 60 d, 90 d and 455 d. The consecutive water drops were considered as slightly simulating the weathering effect.

#### 2.3.2. Hardened Concrete Resistance to Pressurised Water Penetration

The penetration depth of water under pressure EN 12390-8 [49] was conducted on the treated surface of the cylindrical specimens. Three days after applying the treatments, the specimens were exposed to 5 bars (0.5 MPa) hydrostatic pressure (Figure 2) for 72 h and subsequently split in two perpendicularly to the surface receiving the pressurised water to measure penetration depth (Figure 1).

#### 2.3.3. SEM and EDS Analysis

Measurements were performed on a scanning electron microscope type JSM-6980LV (Jeol Ltd., Tokyo, Japan), using AZtec (Aztec SP2, version 4.0; EDS Software of Oxford instruments: High Wycombe, UK, 2018) coupled to EDS-detector type Oxford instrument ultimmax (High Wycombe, UK). A specimen with each of the different applied bioproducts was prepared (specimen size 20 mm diameter and 10 mm high) after use in the water drop absorption assay. The specimens for the tests were selected based on the best results from the water absorption tests, considering the longer absorption times and the homogeneity on the surface of the piece tested. For SEM data acquisition, a large field detector (LFD) and a pressure of 0.3 mbar were chosen. Images were taken at a magnification of 100× and an acceleration voltage of 20 kV. For EDS analysis, the same conditions as SEM data acquisition were used.

## 3. Results and Discussion

### 3.1. Surface Absorption

The mean absorption times for the SP-treated, NSP-treated and reference specimens (in which the bioproduct was replaced by the same volume of water) are graphed in Figure 2. This figure also shows the variation in absorption times of the nine water drop absorption test times.

According to these findings, and in line with previous results of García-González et al. [50], the water repellency generated by the two bioproducts substantially reduced the concrete surface permeability. After 3 days of treatment, the absorption time was reduced by more than 119 times in the SP-treated specimens, and by more than 95 times in the NSP-treated specimens in comparison to the reference. As Figure 3 shows, the efficacy of both bioproducts declined progressively due to the washout effect of the water drop test, and to time itself as in materials exposed to uncontrolled atmospheres with variable weathering agents. This possibility calls for more thorough investigation in future studies. Moreover, although the organic surface treatments applied to protect mortars and concretes are highly eco-friendly and deliver promising results, they have been used and studied less frequently than inorganic treatments due to their short service life [51,52] and scant penetration depth [21].

Even 450 days after treatment and nine water drop tests, the SP specimens exhibited 22 times, and the NSP specimens 9 times, longer absorption times than the water-treated reference specimens. The decline in impermeability over time observed in the latter, from a mean of 39 s in the earliest ages to 33 s in the latest, corroborated the degenerative washout effect of dripping water on the concrete surfaces over time.

It was also possible to observe a difference in behaviour between the two bioproducts, which showed substantially longer uptake times (25% to 93%, depending on the test age) in the concrete treated with the sonicated bioproduct in comparison to the non-sonicated one. The explanation should lay in the nature of the bioproduct, as sonication breaks down the cell membranes of the microorganisms, releasing the intracellular PHAs and other polymers (proteins, carbohydrates, lipids, and so on), and as such conferred a higher effective concentration of bioproducts in the SP than in the NSP specimens.

### 3.2. Energy Dispersive X-ray Analysis (SEM/EDS)

The organic nature of the bioproducts was well reflected by the carbon content in Energy dispersive X-ray analysis, as shown in Table 3 and in Figure 3 and Figure 4, where through mass diffractogram it was possible to detect the elements that composed the surfaces of the SP and NSP specimens. The carbon percentage by weight in the SP was up to 8% higher than in SNP. Figure 4 shows the surface of both specimens, obtained by SEM, on which the carbon layer was overlapped, in the same regions as those analysed in Figure 4. It was possible to observe that despite the multiple washes of the specimen’s surface, due to the waterdrop test, the carbon was practically distributed throughout the surface, being more abundant and homogeneous in the SP specimen than in the NSP specimen. This greater homogeneity can also be seen in Figure 5 (high-resolution images obtained by SEM).

These results correlate with those obtained by the drop absorption test, explaining in greater detail why the sonicated bioproduct is more effective than the non-sonicated one.

### 3.3. Penetration Depth of Water under Pressure

The results in Figure 6 show the depth of the water penetration front under pressure in the SP, NSP and water-treated specimens. The efficacy was similar for both treatments, with SP reducing the penetration depth by 51% and NSP by 53%, relative to the reference. The explanation for the difference observed between those findings and the water drop test is to be found in the protective film formed by the two bioproducts. The resulting barrier limited pressurised water penetration considerably and similarly in the two biotreatments. Due to the difference in pressure between the two tests, the sonicated bioproduct proved to be more effective at normal atmospheric pressure and the non-sonicated compound was slightly more effective at the high pressure used in the penetration-depth test. The latter finding might be attributable to the greater resistance to high pressure exerted by the more compact nucleation sites in the non-sonicated MMC.

In turn, the SP water penetration front was more uniform than the NSP front, denoting a more uniform distribution in the former than in the latter, where the profile was visibly more irregular (Figure 7). That irregularity was likely due to a less-uniform distribution of the bioproduct. The sonicated MMC exhibited denser areas characterised by clusters formed by the aggregation of different polymers and cell debris and consequently higher bioproduct concentration (more resistant to the penetration of water under pressure) and other areas with a lower concentration (less penetration-resistant).

The findings of both tests consequently attested a significant decline in recycled concrete surface permeability when coated with the innovative bioproducts, which also constitutes a physical barrier that hinders the penetration of substances that contribute to shorten concrete durability and service life.

The comparison of the surface waterproofing treatments’ efficacy over time is challenging due to the wide variety of products used to protect concrete surfaces; the various physical factors affecting compound penetration, including density, viscosity, angle of natural repose and surface tension [25]; and the differences in the tests and procedures used to analyse product properties. In one of the few studies on the use of organic surface treatments to enhance cement-based material durability, Chandra et al. [32] reduced the sorptivity in conventional concrete specimens by approximately 80% after soaking the material in a 100% prickly pear extract. In another work, García-González [53] analysed the effectiveness of the protection afforded mortar specimens by two bioproducts, one extracted from *Escherichia coli* BL21 (DE3) cultures and the other derived from the waste biomass of a polyhydroxyalkanoate-producing mixed microbial culture using tap water containing diluted crude glycerol as growth medium. They observed water drop test absorption times for both products to be more than 10-fold longer than the times recorded for the reference specimens.

## 4. Conclusions

The present research findings stand as proof that bioproducts derived from polyhydroxyalkanoates-accumulating microbial mixed culture using pinewood bio-oil were able to enhance the durability of the tested material as:The two applied bioproducts effectively protected the surface of the concrete, attested to by the substantial increase in resistance to water absorption and water penetration;The sonicated biopolymer demonstrated greater effectiveness under normal atmospheric pressures, as shown by the results in the water drop absorption test;SEM/EDS analyses showed a higher carbon concentration associated with the bioproduct, as well as a more uniform distribution in the sonicated bioproduct, in which microbial cell membranes were broken down;The pressure water penetration test confirmed that the distribution of the sonicated bioproduct was more uniform than the non-sonicated one, the latter presenting areas with higher concentration of the bioproduct. The non-sonicated bioproduct was revealed to be more effective at high pressures.

Eco-friendly organic surface treatments can be effectively used to enhance recycled concrete durability, preventing deterioration in concretes exposed to external agents, given their capacity to lower permeability—a foremost indicator of cement-based material durability. This study may inspire other routes for future research on the efficacy of such compounds, the application of which to concrete surfaces would lengthen the material’s service life. Their enhancement of recycled concrete performance, in turn, would encourage the use of recycled CDW aggregates, thereby contributing to a reduction in the volume of such waste to manage and decrease the extraction of raw aggregates.

## Figures and Tables

**Figure 1 materials-14-06545-f001:**
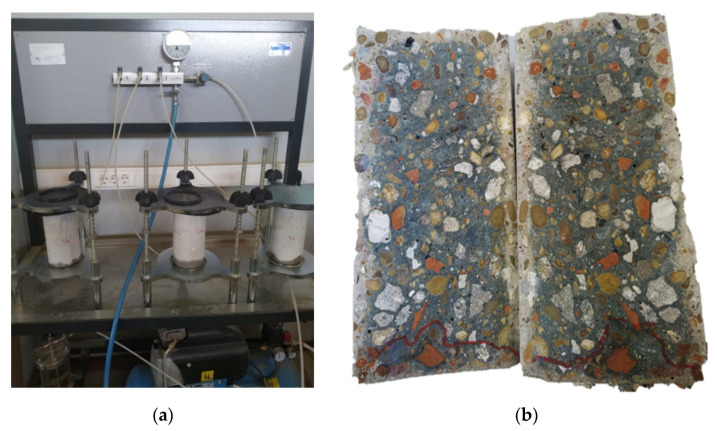
Water penetration under pressure test (**a**) and penetration depth in a specimen (**b**).

**Figure 2 materials-14-06545-f002:**
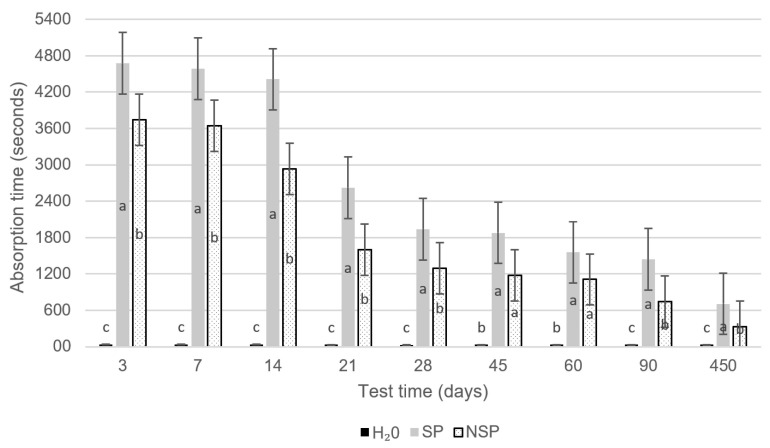
Average water-drop absorption time for sonicated (SP) and non-sonicated (NSP) bioproduct-treated recycled aggregate concrete. Statistically significant results with respect to the control treatment (*p* < 0.05) are indicated with different letters as result of a post-hoc Duncan multi-range test that was run on the data.

**Figure 3 materials-14-06545-f003:**
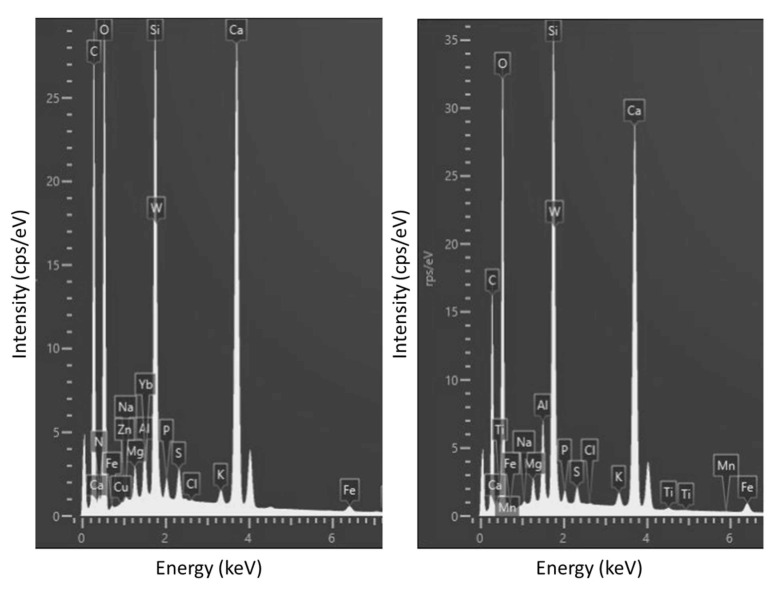
EDS spectra of element content of region 2 on the SP sample on the left and region 1 of the NSP sample on the right.

**Figure 4 materials-14-06545-f004:**
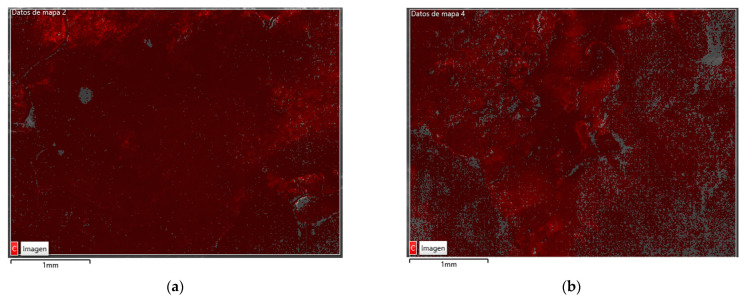
Secondary electron image made with SEM, with a layer superimposed with the carbon distribution of regions 2 SP (**a**) and 1 NSP (**b**).

**Figure 5 materials-14-06545-f005:**
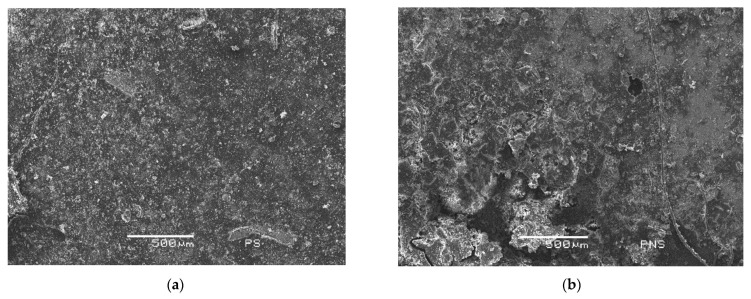
SEM images after 450 days and nine water drop tests. (**a**) an SP sample, and (**b**) an NSP sample.

**Figure 6 materials-14-06545-f006:**
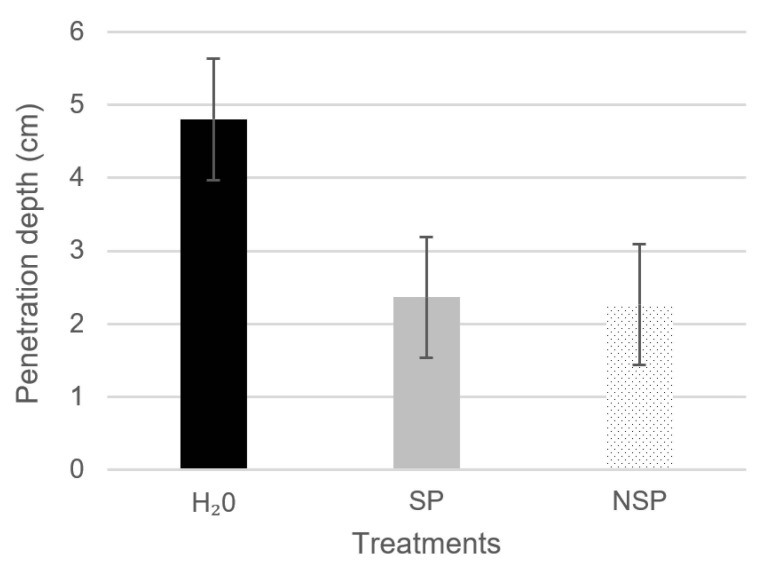
Average penetration depth of water under pressure in water-, SP- and NSP-treated recycled concrete specimens.

**Figure 7 materials-14-06545-f007:**
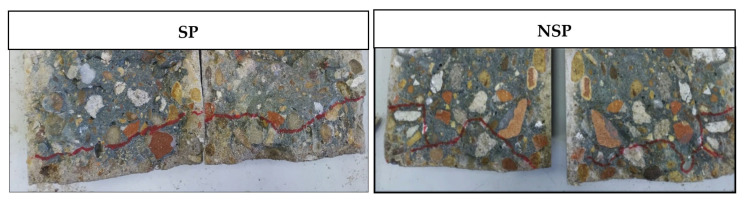
Examples of penetration fronts in SP and NSP specimens after the water under pressure test.

**Table 1 materials-14-06545-t001:** Recycled concrete composition per cubic metre.

Material	Composition
Water	215 L
Cement	391 kg
Sand	716 kg
Gravel	447 kg
MRA	447 kg

**Table 2 materials-14-06545-t002:** Mixed recycled aggregate components.

Component	wt%
Unbound aggregate (natural aggregate with no attached cement mortar)	17.5
Masonry and fired clay (bricks, tiles, stoneware, sanitary ware, etc.)	33.6
Concrete and mortar (natural aggregate with bound cement mortar)	44.1
Asphalt	0.4
Glass	0.8
Calcium sulphates (gypsum)	3.5
Other impurities (wood, paper, metals, plastic, etc.)	0.1

**Table 3 materials-14-06545-t003:** Relative elemental composition of samples’ surfaces analysed with EDS.

	Position	Relative Elemental Content (%)								
		C	O	Ca	Si	Al	Fe	Mg	K	N
NSP	1	27.6	43.9	14.3	9.5	1.6	1.0	0.6	0.4	0
	2	24.1	46.7	14.5	10.1	1.6	0.8	0.6	0.6	0
SP	1	35.2	38.9	12.6	6.8	3.3	0.4	0.6	0.3	3.3
	2	35.8	38.,6	12.5	5.8	1.0	0.5	0.5	0.3	4.0

## Data Availability

The data presented in this study are available on request from the corresponding author.

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
