# Peer review of "Use of Mixed Microbial Cultures to Protect Recycled Concrete Surfaces: A Preliminary Study"

_materials, 2021, doi:10.3390/ma14216545_

Round 1

Reviewer 1 Report

-We live now in a climate emergency so its most strange that the authors have not start the paper by mentioning exactly that. It seems that they are not aware about the words of a Professor of Physics at the University of Oxford authored a paper where one can read the following:

 “Let’s get this on the table right away, without mincing words. With regard to the climate crisis, yes, it’s time to panic”

Pierrehumbert, R., 2019. There is no Plan B for dealing with the climate crisis. Bulletin of the Atomic Scientists, pp.1-7.

So please start the introduction by draw a connection between environmental degradation, resource efficiency, and concrete durability

-The first reference is outdated unless that authors are saying that the concrete production in 2012 is the same of 2021. Also reference 1 is not a study about concrete production but instead about recycled aggregates. So please use a recent study that focus on concrete production, one that can give some estimates about concrete production in future.

 - Reference 2 is used to show statistics of construction and demolition wastes still this reference has nothing to do about construction and demolition wastes.

-Reference 5 comments on the European Directive 2008/98/EC still the year 2020 has already passed and the target was not fulfilled by the European countries. In fact several authors have criticized the European Directive has being ineffective. Arm et al. (2017) states that the Directive is very sensitive to how the legal definitions of waste and recovery are interpreted in the Member States. They also mention that the Directive does not distinguish between the various recovery processes and also that its weight-based approach favours large and heavy waste streams. Gálvez-Martos et al. (2018)

Arm, M., Wik, O., Engelsen, C. J., Erlandsson, M., Hjelmar, O., & Wahlström, M. (2017). How does the European recovery target for construction & demolition waste affect resource management?. Waste and Biomass Valorization8(5), 1491-1504.

Gálvez-Martos, J. L., Styles, D., Schoenberger, H., & Zeschmar-Lahl, B. (2018). Construction and demolition waste best management practice in Europe. Resources, Conservation and Recycling136, 166-178.

-Reference 6 is not the most appropriate to back a statement about concrete durability. See below two references that are more suitable especially in the context of the core of the paper:

Pan, X., Shi, Z., Shi, C., Ling, T. C., & Li, N. (2017). A review on concrete surface treatment Part I: Types and mechanisms. Construction and Building Materials132, 578-590.

Pan, X., Shi, Z., Shi, C., Ling, T. C., & Li, N. (2017). A review on surface treatment for concrete–Part 2: Performance. Construction and Building Materials133, 81-90.

- References 8 and 9 are outdated (2001 and 1997)

- What´s the contamination level of the mixed recycled aggregates ?

- Since cost is extremely important for construction industry what´s the expected cost of the treatment presented in this research ?

Reviewer 2 Report

The article covers the topic of the use of mixed microbial cultures to protect recycled concrete

surfaces. In my opinion, article presents valuable content. The subject and the supporting analysis are informative and present added value to the body of knowledge on the subject area. The topic of the article is in scope of journal. The article is very clearly written and edited.

The assumptions used in the analysis are correct and appropriate at this stage of the analysis. It is hoped that in the future the authors will expand the analysis, particularly in terms of comparison with other techniques for increasing the durability of concrete. The lack of such a comparison is, in my opinion, the biggest disadvantage of the reviewed article. But as the authors themselves noted in the title, the results of preliminary research were presented.

The version of the article sent to me contains some very small editing errors. Conclusions should be formatted as a numbered or bulleted list according to the journal template.

Author Response

The article covers the topic of the use of mixed microbial cultures to protect recycled concrete surfaces. In my opinion, article presents valuable content. The subject and the supporting analysis are informative and present added value to the body of knowledge on the subject area. The topic of the article is in scope of journal. The article is very clearly written and edited.

The assumptions used in the analysis are correct and appropriate at this stage of the analysis. It is hoped that in the future the authors will expand the analysis, particularly in terms of comparison with other techniques for increasing the durability of concrete. The lack of such a comparison is, in my opinion, the biggest disadvantage of the reviewed article. But as the authors themselves noted in the title, the results of preliminary research were presented.

The version of the article sent to me contains some very small editing errors. Conclusions should be formatted as a numbered or bulleted list according to the journal template.

Thank you for the comments. Conclusions are now presented as bulleted list.

Reviewer 3 Report

This work aims to study the capacity of bioproducts derived from polyhydroxyalkanoates-producing mixed microbial cultures to waterproof cement-based materials sufaces for increasing durability.

The manuscript reflects a matured research, however some aspects should be considered and corrected before publication.

  • Introduction: this sentence: "Hence the need to improve the performance of concrete-based materials during their life cycle." seems incomplete.
  • Introduction: A full stop is required afetr this sentence: "...while having an excess of carbon source [22],[23]"
  • Introduction: after the above sentence, "functios" is found and needs to be corrected.
  • 2.3.3. SEM and EDS analisys: revise "seletecd".
  • Water drop absorption test seems to have been done under uncontrolled ambient conditions (humidity, temperature...) and then its results cannot provide conclusive conclusions. Can you provide more insight into the development of this experiment in order to accept its results?
  • Energy Dispersive X-ray Analysis (SEM/EDS): Please revise "bioporduct".

Round 2

Reviewer 3 Report

The manuscript has been revised and modified in order to improve its quality,. According to this reviewer opinion this manuscript deserves publication in Materials Journal.